# PerturbScore: Connecting Discrete and Continuous Perturbations in NLP

**Linyang Li**[*], **Ke Ren,**[*] **Yunfan Shao, Pengyu Wang,**
**Xipeng Qiu** [†]
School of Computer Science, Fudan University
Shanghai Key Laboratory of Intelligent Information Processing, Fudan University
`{kren22,pywang22}@m.fudan.edu.cn`
`{linyangli19, yfshao19, xpqiu}@fudan.edu.cn`

## Abstract

With the rapid development of neural network applications in NLP, model robustness problem is gaining more attention. Different from computer vision, the discrete nature of texts makes it more challenging to explore robustness in NLP. Therefore, in this paper, we aim to connect discrete perturbations with continuous perturbations, therefore we can use such connections as a bridge to help understand discrete perturbations in NLP models. Specifically, we first explore how to connect and measure the correlation between discrete perturbations and continuous perturbations. Then we design a regression task as a PerturbScore to learn the correlation automatically. Through experimental results, we find that we can build a connection between discrete and continuous perturbations and use the proposed PerturbScore to learn such correlation, surpassing previous methods used in discrete perturbation measuring. Further, the proposed PerturbScore can be well generalized to different datasets, perturbation methods, indicating that we can use it as a powerful tool to study model robustness in NLP. [1]

## 1 Introduction

Natural language processing (NLP) applications based on neural networks are developing rapidly, exemplified by applications based on pre-trained models (Devlin et al., 2018) such as ChatGPT [2] (Brown et al., 2020), machine translation systems (Bahdanau et al., 2014), question-answering systems (Rajpurkar et al., 2016). While they are growing at an incredible speed, it is of great concern how we can trust these neural networks. Therefore, exploring model robustness in NLP is essential for future NLP developments. Model

robustness problems mostly focus on exploring the model behavior when the inputs are perturbed. However, unlike the computer vision field, the discrete nature of natural language makes it more challenging to define, construct and measure the perturbations added to the texts.

Previous works usually separate two lines of work concerning discrete perturbations and continuous perturbations: In the computer vision (CV) field, continuous perturbations are widely explored (Goodfellow et al., 2014) since studying perturbations can help improve model robustness (Madry et al., 2019) and generalization abilities (Hendrycks and Dietterich, 2019; Hendrycks et al., 2021). As for perturbations in NLP, the difference and challenge in discrete perturbations constrain the development of model robustness in NLP. Jin et al. (2019); Zang et al. (2020); Li et al. (2020) craft adversarial examples with synonyms as word-level perturbations, which is hard to measure whether the perturbations are imperceptible. Also, crafting these perturbations would face a combinatorial explosion problem. As studying discrete perturbations is more challenging than continuous perturbations, it is intuitive to wonder: *can we find the correlations between discrete and continuous perturbations in NLP and study continuous perturbations instead?*

In this paper, we aim to explore connections between discrete and continuous perturbations in NLP models hoping that such connections can help studies of model robustness in NLP. We first give definitions and measuring standards of perturbations in discrete and continuous space and align the form and notations for studying their correlations. Then we make several assumptions to provide the possibility of connecting discrete and continuous perturbations. Specifically, we assume that discrete perturbations and continuous perturbations have similar effects on neural models when being added to the input to perturb the

---

[*]Equal Contribution.
[†]Corresponding author.

[1]We will release our code and generated datasets at `https://github.com/renke999/PerturbScore`
[2]`https://openai.com/blog/chatgpt/`

model. Therefore, we are able to search for a continuous perturbation that can be considered as a substitution for the discrete perturbation. We then introduce a method to quantify the correlations between discrete and continuous perturbations and design a regression task to automatically learn the correlation. Specifically, we use the gradient-projection descent method to search for a minimum continuous perturbation that has similar effects on the model behavior with the discrete perturbation. After quantifying the correlations between discrete and continuous perturbations, we use an additional neural network named **PerturbScorer** to learn such a correlation. That is, given the original input and a discrete perturbation, we use a neural network to predict its continuous perturbation range.

We construct experiments on IMDB and AG's News datasets, which are widely used datasets in NLP robustness studies. We first explore the correlations between discrete and continuous perturbations when they are used to perturb fine-tuned models such as BERT; then we test the performances of the learned network and empirically verify that the correlations between perturbations can be learned through a neural network, providing evidence for researchers to study continuous perturbations in NLP as a substitute for discrete perturbations.

Further, we design extensive analytical experiments and through the experimental results, we make several non-trivial conclusions: (1) we can build a connection between discrete and continuous perturbations; (2) such a connection can be generalized and help improve model robustness and generalization abilities; (3) continuous-space adversarial training is effective because it narrows the gap between discrete and continuous space.

To summarize, in this paper, we explore the correlations between discrete and continuous perturbation, a fundamental challenge in robustness studies in NLP. We provide detailed notations and make assumptions to explore the correlations between perturbations and propose a method to connect these perturbations; further, we design a PerturbScorer to learn such correlation; through experimental results, we show that we can connect discrete perturbations with continuous perturbations. We are hoping that the concept of studying discrete perturbations in NLP through building the connections between continuous perturbations can provide hints for future studies.

## 2 Related Work

### 2.1 Model Robustness and Perturbations

Robustness problems are widely explored in the deep learning field: Goodfellow et al. (2014); Carlini and Wagner (2016) discussed the possibility of crafting gradient-based perturbations as adversarial examples to mislead neural models. Madry et al. (2019) introduces the projected gradient descent method to construct perturbations. Hendrycks and Dietterich (2019); Hendrycks et al. (2021) discussed more general perturbations such as Gaussian noise, blurs, etc. in the distribution shift scenarios. When the perturbations are continuous, studies focus on exploring connections between model robustness and model accuracy (Zhang et al., 2019a; Yang et al., 2020) and plenty of analytical works dive deep into the model robustness studies (Pinot et al., 2019). These robustness studies assume that the perturbations are continuous, therefore, they are not suitable for discrete perturbations and NLP robustness studies.

### 2.2 Perturbations in NLP

In the NLP field, the robustness problem becomes more challenging due to the discrete nature of texts. Ebrahimi et al. (2017) explores crafting character-level and word-level perturbations as adversarial examples to attack NLP models. Follow-up works such as Jin et al. (2019); Zang et al. (2020); Li et al. (2020) aim to find better methods to craft semantic-preserving adversaries. As for more general perturbations, Jia and Liang (2017) explores how adding random sentences can mislead question-answering systems; Yi et al. (2021) explores how adversarial training improves out-of-distribution model generalization problems. Unlike the continuous perturbations explored in the computer vision field, the NLP field rarely discusses how the model behaves in robustness against adversaries and generalization abilities. Zhu et al. (2019) introduces embedding-space gradient-based adversarial training and discovers that continuous space adversaries can help improve NLP model generalization abilities without further explanation. Li et al. (2021) founds that gradient-based adversarial training can be used in defense against word-substitution attacks. In general, robustness studies in NLP rarely focus on finding the correlation between discrete and continuous perturbations which separate works in vision and language fields.

## 3 Connecting Perturbations

### 3.1 Defining Perturbations

We first define perturbations in deep neural networks for NLP applications:

Given an input text $S = [w_0, w_1, \cdots, w_n, \cdots]$, the corresponding embedding of $S$ is $X = [\vec{x_0}, \vec{x_1}, \cdots, \vec{x_n}, \cdots]$. The prediction of the input $S$ is denoted as $f(X)$. Here, we use the embedding output $X$ as the model input since we aim to connect the discrete perturbations with continuous perturbations in the embedding space. When the input text is maliciously attacked or perturbed by random noise, the input text becomes $S'$. We use $\mathcal{P}(S)$ to denote the perturbation process therefore the perturbed text $S' = S + \mathcal{P}(S)$. The perturbation function $\mathcal{P}(S)$ can be various methods including adversarial attacks and random perturbations. Representative adversarial attack methods are word-substitution adversarial attacks such as HotFlip (Ebrahimi et al., 2017), Textfooler (Jin et al., 2019) and BERT-Attack (Li et al., 2020). Unlike random perturbations such as random deleting or replacing words/characters, adversarial attack methods aim to find the minimum amount of character- or word-level substitutions that can mislead target models.

Unlike in the computer vision field where continuous perturbations can be directly added to the input, the continuous perturbations can only be added to the embedding output $X$ in the language field. For embedding output $X \in \mathbb{R}^{l*d}$ with sequence length $l$ and hidden size $d$, we have perturbed output $X' = X + \boldsymbol{\delta}$. The continuous perturbation $\boldsymbol{\delta} \in \mathbb{R}^{l*d}$ can be random noise (Hendrycks and Dietterich, 2019) such as Gaussian noise, blurs, pixelate, or adversarial perturbations. A representative method to generate adversarial perturbation $\boldsymbol{\delta}$ is the Fast Gradient Sign Method (FGSM) (Goodfellow et al., 2014). Given a target model $f_\theta(\cdot)$, the perturbation of sample $S$ is generated based on the gradients: $\boldsymbol{\delta} = \alpha \cdot \text{sgn}(\nabla_X f_\theta(X, y))$. Here, $\alpha$ is a hyper-parameter controlling the perturbation range.

### 3.2 Measuring Perturbations

After defining perturbations, it is important to measure how perturbations affect neural models.

Measuring the severity of the perturbation is a challenge in discrete text perturbations. The similarity between the perturbed and original texts cannot be easily measured. We use $\mathcal{A}(\mathcal{P}(S))$ to measure the perturbation intensity of perturbation $\mathcal{P}(S)$ added to the original text $S$, which could be edit-distance, semantic shift, grammar change, etc. For instance, when we use edit-distance as $\mathcal{A}(\mathcal{P}(S))$, we assume that the fewer tokens the original text is replaced, the less the text is perturbed. Besides edit-distance, Jin et al. (2019) introduces USE (Cer et al., 2018) to measure the perturbation intensity. We assume that the less semantic information is changed, the less the text is perturbed. In general, finding an accurate measure strategy $\mathcal{A}(\cdot)$ is challenging since the standard can be diversified and subjective when measuring discrete perturbations.

On the other hand, continuous perturbations can be measured by constraining the $\ell_p$-norm $||\boldsymbol{\delta}||_p$ of the perturbations $\boldsymbol{\delta}$. The most common constraint is the $\ell_2$-norm. Compared with measuring discrete perturbation, it is easy and straightforward to measure the continuous perturbation range.

### 3.3 Connecting Perturbations

As illustrated, it is challenging to measure discrete perturbations, which makes it more difficult to explore how perturbations affect the model behavior. Meanwhile, another challenge is that constructing discrete perturbations is also challenging, for instance, replacing discrete tokens in a multi-token text with multiple candidates for each token is a combinatorial explosion problem. Therefore, since measuring and studying continuous perturbations is more convenient, instead of searching for methods to evaluate the shift caused by discrete perturbations, we aim to build a connection between discrete perturbations and continuous perturbations and explore how the continuous perturbation affects model behaviors instead. We hope that by connecting discrete perturbations to continuous perturbations, we can introduce new perspectives to NLP field model robustness problems.

To build the connection between discrete perturbations and continuous perturbations, we make several assumptions:

***Assumption*** **1.** Continuous perturbations $\boldsymbol{\delta}$ added to $X$ have similar effects on target model $f_\theta$ compared to discrete perturbations added to $S$. That is, both types of perturbations can cause a model prediction shift, and stronger perturbations would cause more damage to the model.

| Texts $S$ from AG's News | Perturbations $\mathcal{P}(S)$ | Perturbation Measure $\mathcal{A}(\mathcal{P}(S), S)$ | Outputs Shift $f_\theta(S) \to f_\theta(S + \mathcal{P}(S))$ |
|---|---|---|---|
| Apple Recalls Batch of PowerBook Batteries: Apple, in cooperation with the US Consumer Product Safety Commission said it would voluntarily recall about 28,000 rechargeable batteries used in its 15-inch PowerBook G4 notebooks. | ●Textfooler: Apple -> Mitt Batch -> Afar cooperation -> cooperatives Product -> Commodities recall -> reminds | edit-distance:5 USE: 0.889 BERTScore: 0.951 | $f_\theta(\cdot)$: BERT Sci/Tech (100%) -> Business (81%) |
| Apple Recalls Batch of PowerBook Batteries: Apple, in cooperation with the US Consumer Product Safety Commission said it would voluntarily recall about 28,000 rechargeable batteries used in its 15-inch PowerBook G4 notebooks. | ■Random Perturbation : Apple -> Overcast cooperation -> striker Consumer -> Concessions said -> rewarded voluntarily -> trenton | edit-distance:5 USE: 0.880 BERTScore: 0.947 | $f_\theta(\cdot)$: BERT Sci/Tech (100%) -> Sci/Tech (90%) |

Table 1: Selected samples with same edit-distance perturbations showing discrete perturbation constructions and measurements of discrete perturbations. The model output shifts are different from edit-distance or BERTScore.

***Assumption 2.*** Target model $f_\theta$ follows Lipschitz constraint: when $||\boldsymbol{\delta}||_2 < \epsilon$, $||f_\theta(X + \boldsymbol{\delta}) - f_\theta(X)||_2 < K \cdot \epsilon$. Here, we assume that in NLP models, such as a fine-tuned BERT, small perturbations in the embedding space do not cause severe damage to model outputs. Otherwise, the behavior change caused by input perturbations is hard to predict, and finding correlations between perturbations is more challenging.

***Assumption 3.*** For a discrete perturbation $\mathcal{P}(S)$, there exist a continuous perturbation $\boldsymbol{\delta}$ that satisfies: $\epsilon - \varepsilon < ||\boldsymbol{\delta}||_2 < \epsilon$, here, $\varepsilon$ is a small interval. And such $\boldsymbol{\delta}$ satisfies:

$$\left| \frac{||f_\theta(S + \mathcal{P}(S)) - f_\theta(S)||_2}{||f_\theta(S + \mathcal{P}(S))||_2 \cdot ||f_\theta(S)||_2} - \frac{||f_\theta(X + \boldsymbol{\delta}) - f_\theta(X)||_2}{||f_\theta(X + \boldsymbol{\delta})||_2 \cdot ||f_\theta(X)||_2} \right| < \phi$$

, here, $\phi$ is a hyper-parameter, and for simplification, $f_\theta(\cdot)$ takes both discrete tokens $S$ and embedding output $X$ of the discrete tokens $S$ as input, skipping the embedding process.

We assume that the continuous perturbation has a similar effect on model $f_\theta(\cdot)$, therefore, when the absolute value of the gap between model shift caused by discrete and continuous perturbations is small, we consider they are equal in perturbing neural models. Therefore, we can use continuous perturbations as an approximation of discrete perturbations by building connections between them.

### 3.4 Quantify Connections

After assuming that we can build connections between discrete and continuous perturbations, we aim to quantify such connections. For a discrete perturbation $\mathcal{P}(S)$, we find the minimum continuous perturbation $\delta$ under *Assumption* 3. Specifically, we aim to find the proper norm-bound

$\epsilon$ that under such a norm-bound, there exists a perturbation $\boldsymbol{\delta}$ satisfies *Assumption* 3 mentioned above. Therefore, when $S$ and $\mathcal{P}(S)$ is fixed, the goal is to find a norm-bound $\epsilon$:

$$\arg \min_\epsilon \max_{||\boldsymbol{\delta}||_2 < \epsilon} \left( \frac{||f_\theta(X + \boldsymbol{\delta}) - f_\theta(X)||_2}{||f_\theta(X + \boldsymbol{\delta})||_2 \cdot ||f_\theta(X)||_2} \right) \quad (1)$$

Therefore, we would obtain a data tuple $[S, \mathcal{P}(S), \epsilon]$, which is the correlation of discrete and continuous perturbations. We are hoping that we can empirically verify that the data tuple can be connected, and verify the assumptions made above.

---

**Algorithm 1** Obtaining norm-bound $\epsilon$

---

**Require:** Inputs $X, S, \mathcal{P}(S)$, label $y$, search step $T_a$, norm range interval $\varepsilon$

1: $\Gamma \leftarrow \frac{||f_\theta(S + \mathcal{P}(S)) - f_\theta(S)||_2}{||f_\theta(S + \mathcal{P}(S))||_2 \cdot ||f_\theta(S)||_2}$
2: **for** $\epsilon = 0, \varepsilon, 2\varepsilon, ...$ **do**
3: $\quad \boldsymbol{\delta}_0 \leftarrow 0$
4: $\quad$ **for** $t = 0, 1, ...T_a$ **do**
5: $\quad\quad \boldsymbol{g_\delta} \leftarrow \triangledown_\delta \mathcal{L}(f_\theta(X + \boldsymbol{\delta}_t), y)$
6: $\quad\quad$ // Get Gradients
7: $\quad\quad \boldsymbol{\delta}_t \leftarrow \prod_{||\boldsymbol{\delta}||_2 < \epsilon} (\boldsymbol{\delta}_t + \alpha \cdot \frac{\boldsymbol{g_\delta}}{||\boldsymbol{g_\delta}||_2})$
8: $\quad\quad$ // Get Perturbation
9: $\quad\quad$ **if** $\left| \frac{||f_\theta(X + \boldsymbol{\delta}_t) - f_\theta(X)||_2}{||f_\theta(X + \boldsymbol{\delta}_t)||_2 \cdot ||f_\theta(X)||_2} - \Gamma \right| < \phi$ **then**
10: $\quad\quad\quad$ return tuple $[S, \mathcal{P}(S), \epsilon]$
11: $\quad\quad$ **else**
12: $\quad\quad\quad$ discard tuple

---

In practice, to obtain $\epsilon$, we use a standard projected-gradient-descent (PGD) (Madry et al., 2019) method. As seen in Algorithm 1, we use multi-step gradient-descent to generate perturbations within the range $\epsilon$, which is the perturbation generation used in gradient-based

adversarial training. Specifically, the notation $x$ used in line 7 is to constrain the perturbations within the norm bound $\epsilon$. In line 9, we pick the $\epsilon$ that satisfies the assumption that there exists a continuous perturbation that has a similar effect on neural models compared with discrete perturbations. Therefore, once the perturbation $\boldsymbol{\delta}$ obtains ideal effects on the neural model (bigger than the effects caused by discrete perturbations), we consider the $\epsilon$ found is the proper one. A special case is that if the continuous perturbation effect $\frac{||f_\theta(X+\boldsymbol{\delta})-f_\theta(X)||_2}{||f_\theta(X+\boldsymbol{\delta})||_2 \cdot ||f_\theta(X)||_2}$ is way bigger than $\Gamma$, we simply drop the sample.

### 3.5 PerturbScorer

After constructing the quantification of correlations between discrete and continuous perturbations, we design a PerturbScorer to score the correlations.

The perturbation $\boldsymbol{\delta}$ is a continuous variant, therefore, we formulate a regression task to learn the range of $\epsilon$ given $S$ and $\mathcal{P}(S)$. We train the task as the PerturbScorer $\mathcal{M}([S, \mathcal{P}(S)], \epsilon)$ to measure the correlation between discrete and continuous perturbations in the target model $f_\theta(\cdot)$.

Considering that the perturbation $\mathcal{P}(S)$ should be a small perturbation, we use a simple strategy that concatenates original texts and perturbations in $\mathcal{P}(S)$ as the input of the regression task to learn the correlation. We simply concat the perturbations behind the original token, (e.g.: ..., it would recall [reminds] ... ). Such patterns help the model understand the perturbation of the original texts. Then we use the crafted inputs to predict the norm bounds of the correlated continuous perturbations.

## 4 Experiments

### 4.1 Dataset Construction

To explore the correlations between discrete and continuous perturbations, we use several datasets widely used in exploring model robustness in NLP. We use the IMDB dataset (Maas et al., 2011) and the AG's News dataset (Zhang et al., 2015) which are text classification tasks with an average text length of 220 and 47 accordingly.

We use two widely used perturbation methods $\mathcal{P}(S)$, Textfooler (Jin et al., 2019) and random-perturb. In the Textfooler method, we follow the standard generation process and save perturbations in multiple queries regardless of the attack result, which is different from its original usage that keeps finding perturbations until the attack is successful.

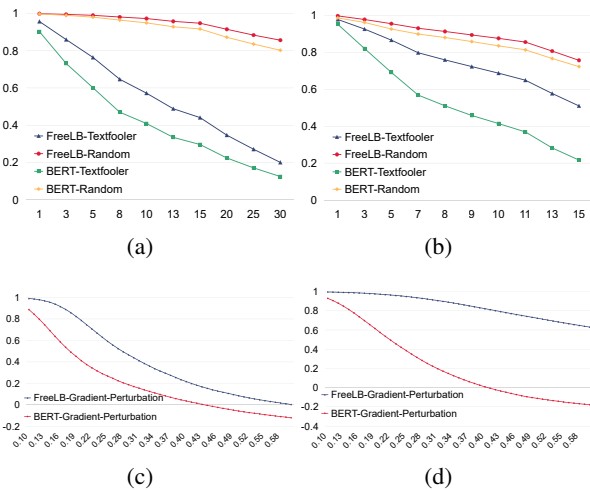

Figure 1: Average Model Shift of Discrete and Continuous Perturbations. (a) is the curve of $\frac{||f_\theta(S+\mathcal{P}(S))-f_\theta(S)||_2}{||f_\theta(S+\mathcal{P}(S))||_2 \cdot ||f_\theta(S)||_2}$ and edit-distance, containing curves of neural model $f_\theta$ including FreeLB-trained model and BERT fine-tuned model and perturbation method $\mathcal{P}(S)$ including Textfooler and Random perturbation tested on the IMDB dataset; (c) is the curve of $\frac{||f_\theta(X+\boldsymbol{\delta}_t)-f_\theta(X)||_2}{||f_\theta(X+\boldsymbol{\delta}_t)||_2 \cdot ||f_\theta(X)||_2}$ and norm-ball range, containing curves of FreeLB-trained model and BERT-fine-tuned model tested on the IMDB dataset; (b) and (d) are the corresponding results of the AG's News dataset.

For each input text $S$, we have multiple $\mathcal{P}(S)$ with different edit-distance differences.

In the random perturb method, we randomly replace a token using a random word from a general vocabulary, which is the vocabulary used to obtain synonyms in the Textfooler method (Mrkšić et al., 2016; Jin et al., 2019). Similar to the Textfooler perturbation method, we also collect multiple perturbations per text including different numbers, places of substitutes, and different substitutes.

In generating continuous perturbations, we use the PGD method to find the minimum continuous perturbation that has a similar model prediction shift compared to the discrete perturbation. Specifically, we set the adversarial step $T_a = 15$ and the adversarial learning rate $\alpha = 1e-1$, the norm-ball search interval $\varepsilon$ of $\epsilon$ is set to 0.01 and the discard parameter $\phi$ is set to 0.005.

In Figure 1, we draw curves exploring the connection between model output shifts and different levels of perturbations. As shown, we can observe that the average model output shifts caused by discrete and continuous perturbations show consistency with edit distance and norm-ball range. We observe that when the perturbations grow

| Range of $\epsilon$ | | | [0,1e-1] | (0.1, 0.2] | (0.2, 0.3] | (0.3, 0.4] | (0.4, 0.5] | (0.5, 0.6] | (0.6, 1) | TOTAL | Discarded |
|---|---|---|---|---|---|---|---|---|---|---|---|
| Dataset | $\mathcal{P}(S)$ | $f_\theta(\cdot)$ | | | | | | | | | |
| **IMDB** | ■**Rand. Perturb** | ▲BERT | 5563 | 2562 | 613 | 164 | 68 | 33 | 17 | 9020 | 1209 |
| | | ▶FreeLB | 4630 | 2265 | 1075 | 528 | 224 | 104 | 86 | 8912 | 1044 |
| | ●**Textfooler** | ▲BERT | 1850 | 2788 | 2256 | 937 | 327 | 155 | 65 | 8378 | 1273 |
| | | ▶FreeLB | 2004 | 1634 | 1990 | 1579 | 826 | 386 | 256 | 8675 | 1459 |
| **AG's News** | ■**Rand. Perturb** | ▲BERT | 3889 | 3940 | 1011 | 200 | 47 | 24 | 15 | 9126 | 2126 |
| | | ▶FreeLB | 1386 | 1426 | 2199 | 1932 | 1204 | 655 | 634 | 9421 | 794 |
| | ●**Textfooler** | ▲BERT | 1499 | 2792 | 2365 | 983 | 368 | 169 | 77 | 8253 | 2800 |
| | | ▶FreeLB | 928 | 772 | 1420 | 1639 | 1425 | 916 | 1340 | 8440 | 1012 |

Table 2: Statistics of the pair number of constructed correlations between discrete and continuous perturbations.

larger, both discrete and continuous perturbations will cause more damage to neural models. Plus, the models trained by the FreeLB method show better resistance against both discrete and continuous perturbations. These results verify *Assumption* 1 and show that discrete and continuous perturbation can be correlated.

In Table 2, we count the tuple number of different $\epsilon$ ranges in the constructed data tuple of multiple datasets to show the connection between discrete and continuous perturbations. We observe that we only discard a small proportion of collected data tuples, proving that we can successfully find norm ball $\epsilon$ that satisfies *Assumption* 3 that such a continuous perturbation $\delta$ is equivalent to the discrete perturbations $\mathcal{P}(S)$ in interfering neural models $f_\theta(\cdot)$. We also observe that as the discrete perturbation $\mathcal{P}(S)$ is uniformly distributed in the edit-distance range from 1 to 30 in the IMDb Dataset and 1 to 15 in the AG's News Dataset, the continuous perturbations mostly fall in the range that $\epsilon < 2e-1$, indicating that most discrete perturbations with different edit-distances (indicating different numbers of substitutions) only compares to a minimum continuous perturbation. Therefore, learning the correlation between these discrete and continuous perturbations can help understand the discrete perturbations. Further, compared with the random perturbation, the Textfooler method generates more discrete perturbations that have more damage to model predictions and the corresponding continuous perturbations require larger norm balls, indicating that stronger discrete perturbations equal to larger continuous perturbations, providing the possibility to connect discrete perturbations with continuous perturbations.

For the collected data, we select 80% data tuples as the training set and 20 % as the test set in training and testing the PerturbScorer.

## 4.2 Evaluating Quantification of Correlation

After constructing the discrete perturbations and finding the corresponding $\epsilon$ of these discrete perturbations, we are able to explore whether the discrete and continuous perturbations can be connected and show similar effects on neural networks. To evaluate the quantification process of correlations illustrated in Sec. 3.4, we use Kendall and Pearson correlation coefficient index to measure whether the discrete perturbations and the continuous perturbations can be connected.

The goal is to measure the correlation coefficient index such as Kendall and Pearson index between $\mathcal{A}(\mathcal{P}(S))$ and the selected norm-bound $\epsilon$. If the correlation between $\mathcal{A}(\mathcal{P}(S))$ and $\epsilon$ is large, we can verify *Assumption* 3 that assumes there exists a continuous perturbation that equals to the discrete perturbation in interfering neural models. We use several simple $\mathcal{A}(\cdot)$ including edit-distances, BERTScore (Zhang et al., 2019b) and USE (Cer et al., 2018). Here, BERTScore and USE measure the similarity between two sentences, which is reversed compared with edit-distance and perturbation scorer, therefore, we use the opposite number of the BERTScore and USE score as $\mathcal{A}(\cdot)$ to measure the correlation coefficient index.

Further, we can directly measure the correlation coefficient index between the predicted and the found $\epsilon$, exploring whether the PerturbScorer can learn the connection between perturbations, which supports the assumptions we made above in Sec. 3.3 and provides a powerful tool to quantify the discrete perturbations for robustness studies in NLP.

## 4.3 PerturbScorer Training

The training process of the PerturbScorer follows the standard fine-tuning process used in fine-tuning regression tasks such as the STS-B (Cer et al., 2017) dataset using huggingface Transformers

| Method Metric | | | | Edit-Distance Kendall | Spearmann | BERTScore Kendall | Spearmann | USE Kendall | Spearmann | PerturbScorer Kendall | Spearmann |
|---|---|---|---|---|---|---|---|---|---|---|---|
| Dataset | $\mathcal{P}(S)$ | $f_\theta(\cdot)$ | | | | | | | | | |
| **IMDB** | ■**Rand. Perturb** | ▲BERT | | 0.4891 | 0.635 | 0.5952 | 0.7745 | 0.577 | 0.755 | **0.7092** | **0.8648** |
| | | ▶FreeLB | | 0.4921 | 0.6278 | 0.599 | 0.7746 | 0.5753 | 0.7489 | **0.6962** | **0.8401** |
| | ●**Textfooler** | ▲BERT | | 0.5153 | 0.6683 | 0.5497 | 0.7302 | 0.4932 | 0.6697 | **0.7975** | **0.9342** |
| | | ▶FreeLB | | 0.5198 | 0.677 | 0.5532 | 0.7404 | 0.4932 | 0.6735 | **0.8173** | **0.9453** |
| **AG's News** | ■**Rand. Perturb** | ▲BERT | | 0.4377 | 0.5785 | 0.4652 | 0.6352 | 0.4372 | 0.6031 | **0.7647** | **0.9141** |
| | | ▶FreeLB | | 0.5128 | 0.669 | 0.5063 | 0.6845 | 0.4726 | 0.6491 | **0.8117** | **0.9445** |
| | ●**Textfooler** | ▲BERT | | 0.5055 | 0.6585 | 0.5279 | 0.7166 | 0.5013 | 0.6851 | **0.821** | **0.9477** |
| | | ▶FreeLB | | 0.4479 | 0.5923 | 0.4662 | 0.6419 | 0.4489 | 0.6194 | **0.8295** | **0.9533** |

Table 3: PerturbScorer evaluation results and correlation comparison with evaluators including BERTScore, USE, and Edit-Distance.

(Wolf et al., 2019). We set the learning rate to 5e-5 with batch size set to 64 and 128 for IMDB and AG's News datasets and use 4xNvidia 3090 GPUs to run the PerturbScorer training process.

## 4.4 Correlation Quantification Results

In Table 3, we list the correlation coefficient index of different measuring methods of perturbations and the PerturbScorer learned correlation of the perturbations:

We can observe that when we use scorers $\mathcal{A}(\cdot)$ to measure the discrete perturbation, the correlation quantification results between the $\mathcal{A}(\cdot)$ and the obtained continuous perturbation bound is not significant. In different setups including different datasets and perturbation methods, the Kendall correlation is smaller than 0.6 and the Spearman correlation is smaller than 0.8, indicating that the discrete perturbation measuring methods do not have close correlations with the continuous perturbations that have a similar effect to neural models, further proving that these measuring methods cannot properly measure the damage to neural models.

On the other hand, we can observe that when we use the PerturbScorer $\mathcal{M}(\cdot)$ to predict the corresponding continuous perturbations, the correlation scores are significant enough to prove that the PerturbScorer can learn the connection between the discrete perturbations and the continuous perturbations. Such results show that we can use our proposed PerturbScorer as a powerful tool to build a connection between the discrete and continuous perturbations.

## 4.5 Analysis

As we first make assumptions about the correlations between discrete and continuous perturbations, we construct the data tuples and design a PerturbScorer to explore whether the correlation can be learned and generalized by neural networks. By exploring the correlations, we can obtain non-trivial observations that can be helpful in model robustness in NLP:

### 4.5.1 Lipschitz Constraint Tightness

As we observe in Figure 1, the model shift is in direct proportion to the perturbation range, indicating that the target model $f_\theta$ follows a Lipschitz constraint on a general scale. Further, as seen in Table 3, compared with the model trained by the FreeLB method, it is more challenging to study the correlation of the normal fine-tuned BERT as $f_\theta(\cdot)$, indicating that gradient-based adversarial training helps build a tighter connection between discrete and continuous perturbations, providing a perspective to explain why gradient-based adversarial training helps in improving robustness performances and generalization performances in NLP tasks with discrete inputs (Zhu et al., 2019; Li et al., 2021).

### 4.5.2 PerturbScorer Generalization

In Table 3, we show that the correlation between discrete perturbation $\mathcal{P}(S)$ and continuous perturbations range $\epsilon$ of model $f_\theta(\cdot)$ can be learned by a PerturbScorer $\mathcal{M}(\cdot)$, further, we aim to explore whether building such a correlation can be applied to various scenarios in robustness studies in NLP. That is, we explore the generalization ability of PerturbScorer $\mathcal{M}(\cdot)$. We explore whether the learned PerturbScorer $\mathcal{M}(\cdot)$ based on target model $f_\theta(\cdot)$ can be generalized to cross-dataset, cross-perturbation method $\mathcal{P}(S)$, cross-model $f_\theta(\cdot)$, therefore, the application of the PerturbScorer and the concept of learning the correlation between discrete and continuous perturbations can be used in various scenarios. We list thorough results in the Appendix.

| PerturbScorer Testing Setup | | | PerturbScorer Training Setup | | | Edit-Distance | | PerturbScorer | |
|---|---|---|---|---|---|---|---|---|---|
| Dataset | $\mathcal{P}(S)$ | $f_\theta(\cdot)$ | Dataset | $\mathcal{P}(S)$ | $f_\theta(\cdot)$ | Kendall | Spearmann | Kendall | Spearmann |
| IMDB | ■Rand. Perturb | ▲BERT | AG's News | ■Rand. Perturb | ▲BERT | 0.4891 | 0.635 | 0.4757 | 0.6441 |
| | | | IMDB | ●Textfooler | ▲BERT | | | 0.5767 | 0.7532 |
| | | | IMDB | ■Rand. Perturb | ▶FreeLB | | | 0.6192 | 0.7943 |
| IMDB & AG's News | ■Rand. Perturb | ▲BERT | IMDB & AG's News | ■Rand. Perturb | ▲BERT | 0.4053 | 0.5035 | 0.7558 | 0.9047 |
| IMDB | ■Rand. Perturb & ●Textfooler | ▲BERT | IMDB | ■Rand. Perturb & ●Textfooler | ▲BERT | 0.4406 | 0.5838 | 0.7802 | 0.9108 |

Table 4: PerturbScorer generalization tests on cross dataset, perturbation type and model. We also test a combined PerturbScorer trained with multi-perturbations, multi-datasets and multi-model generated data tuples.

We can explore how PerturbScorer $\mathcal{M}(\cdot)$ performs on different datasets or faces different types of perturbations:

**Cross-Perturbation PerturbScorer** In cross-perturbation tests, we observe that when we train the PerturbScorer with random perturbations as $\mathcal{P}(S)$, the PerturbScorer can learn perturbations generated by textfooler, while textfooler-generated perturbations cannot be well generalized. Such results show that we can collect multiple types of perturbations to train a PerturbScorer that can be generalized to recognize various discrete perturbations as a powerful tool to quantify how the discrete perturbations affect neural models.

**Cross-Dataset PerturbScorer** In cross-dataset tests, we observe that when we test the AG's News data tuples using the PerturbScorer trained with the IMDB dataset, the correlation is weakened but still stronger than correlations with edit distance, indicating that the PerturbScorer we train can be generalized to different datasets, showing that the connection between discrete and continuous perturbations is strong in general NLP systems, which provides possibilities of using such correlations in various NLP robustness scenarios.

**Cross-Model PerturbScorer** In general, the correlation between discrete and continuous perturbations is dependent on the neural model $f_\theta(\cdot)$ since the perturbation range $\epsilon$ is calculated based on a certain model $f_\theta(\cdot)$. However, when we test the generalization ability between different neural models $f_\theta(\cdot)$, we observe that the correlation is still close. Therefore, it is possible to build a more general PerturbScorer as a general metric to score the discrete perturbations.

**Combined Scorer** We further build a combined PerturbScorer that is trained by a mixture of data tuples including different datasets, perturbations methods and neural models to explore a more generalized scenario.

As seen in Table 4, when we train a model using mixed data collected, we can build a general PerturbScorer that can successfully predict the correlations between discrete perturbations and the corresponding continuous perturbation ranges. Such a result shows that it is possible to build a general PerturbScorer that can be used in solving different datasets and perturbation types, showing that we can use continuous perturbations as a proxy for discrete perturbations when studying NLP robustness problems.

## 5 Conclusion and Future Directions

In this paper, we focus on a fundamental problem in robustness studies in NLP, which is the discrete nature of texts. The discrete nature isolates NLP robustness studies from well-studied machine learning fields, therefore, we introduce the concept of building connections between discrete and continuous perturbations as a new perspective to explore NLP robustness. We build a PerturbScorer to learn the correlation between discrete and continuous perturbations and find that such a PerturbScorer can learn the connection between perturbations, allowing us to use continuous perturbation ranges as a proxy constraint of discrete perturbations, which avoids the challenge that discrete perturbations are hard to measure. Further, we find that our proposed PerturbScorer can be generalized to different datasets and perturbation methods, indicating that such a process can be further applied in the future in NLP robustness studies. For future directions, we aim to explore more effective methods to build a stronger PerturbScorer and to explore more broad scenarios to utilize the proposed PerturbScorer.

## Limitations

In this work, we explore the discrete perturbation in robustness studies in NLP. We aim to find correlations between discrete perturbations and continuous perturbations since continuous perturbations are easily measured and well-studies in the computer vision field. Our work makes assumptions that discrete perturbations show similar effect to neural networks compared with continuous perturbations, therefore, one limitation of such assumptions is that similar effect does strictly make two types of perturbations equal in nature. We find one perspective to connect the discrete perturbations and continuous perturbations, which is not the only solution. Future works can explore more strict constraints and find stronger connections between discrete and continuous perturbations.

Also, better PerturbScorer designing and the applications based on correlations between discrete and perturbations and PerturbScorers can be further explored. We focus on defining and building the connection between discrete and continuous perturbations, and we do not explore further applications based on these connections and our proposed PerturbScorer. For instance, as the PerturbScorer can be used in scoring the discrete perturbations, it can be used in recognizing differences between sentences or measuring distribution shifts. Also, previous works explore robustness and generalization trade-offs and explainable robustness theories on continuous space, mostly in the computer vision field, our works reveal the potential to explore these problems in NLP, which can be explored in future works.

Further, as large language models (LLMs) are drawing much attention in the NLP community, how strong LLMs behave in connecting discrete and continuous space perturbations remains unexplored, especially when GPT-4 (OpenAI, 2023) is known to support images and texts. As these models are not open-source to the public, we leave exploring the perturbation in LLMs in future works.

## Acknowledgements

This work was supported by the National Natural Science Foundation of China (No. 62236004 and No. 62022027). We would like to extend our gratitude to the anonymous reviewers for their valuable comments. Additionally, we sincerely thank Qipeng Guo for his valuable discussions and insightful suggestions on this study.

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

| PerturbScorer Testing Setup | | | PerturbScorer Training Setup | | | Edit-Distance | | PerturbScorer | |
|---|---|---|---|---|---|---|---|---|---|
| Dataset | $\mathcal{P}(S)$ | $f_\theta(\cdot)$ | Dataset | $\mathcal{P}(S)$ | $f_\theta(\cdot)$ | Kendall | Spearmann | Kendall | Spearmann |
| **Cross-Dataset** | | | | | | | | | |
| IMDB | Rand. Perturb | BERT | AG's News | Rand. Perturb | BERT | 0.4891 | 0.635 | 0.4757 | 0.6441 |
| | | FreeLB | | | FreeLB | 0.4921 | 0.6278 | 0.5449 | 0.715 |
| | Textfooler | BERT | | Textfooler | BERT | 0.5153 | 0.6683 | 0.4607 | 0.6384 |
| | | FreeLB | | | FreeLB | 0.5198 | 0.677 | 0.4873 | 0.666 |
| **Cross-Perturbation** | | | | | | | | | |
| IMDB | Rand. Perturb | BERT | IMDB | Textfooler | BERT | 0.4891 | 0.635 | 0.5767 | 0.7532 |
| | | FreeLB | | | FreeLB | 0.4921 | 0.6278 | 0.5608 | 0.7334 |
| | Textfooler | BERT | | Rand. Perturb | BERT | 0.5153 | 0.6683 | 0.4779 | 0.6543 |
| | | FreeLB | | | FreeLB | 0.5198 | 0.677 | 0.4785 | 0.6545 |
| **Cross-Model** | | | | | | | | | |
| IMDB | Rand. Perturb | BERT | IMDB | Rand. Perturb | FreeLB | 0.4891 | 0.635 | 0.6192 | 0.7943 |
| | | FreeLB | | | BERT | 0.4921 | 0.6278 | 0.5963 | 0.7616 |
| | Textfooler | BERT | | Textfooler | FreeLB | 0.5153 | 0.6683 | 0.5901 | 0.7753 |
| | | FreeLB | | | BERT | 0.5198 | 0.677 | 0.5939 | 0.7819 |
| **Combined-Dataset PerturbScorer** | | | | | | | | | |
| IMDB & AG's News | Rand. Perturb | BERT | IMDB & AG's News | Rand. Perturb | BERT | 0.4053 | 0.5035 | 0.7558 | 0.9047 |
| | | FreeLB | | | FreeLB | 0.2981 | 0.4059 | 0.8013 | 0.9325 |
| | Textfooler | BERT | | Textfooler | BERT | 0.4713 | 0.6267 | 0.8114 | 0.9429 |
| | | FreeLB | | | FreeLB | 0.3548 | 0.4917 | 0.8325 | 0.9549 |
| **Combined-Perturbation PerturbScorer** | | | | | | | | | |
| IMDB | Rand. Perturb & | BERT | IMDB | Rand. Perturb & | BERT | 0.4406 | 0.5838 | 0.7802 | 0.9198 |
| | | FreeLB | | | FreeLB | 0.4539 | 0.5942 | 0.7795 | 0.914 |
| AG's News | Textfooler | BERT | AG's News | Textfooler | BERT | 0.4179 | 0.5579 | 0.803 | 0.9378 |
| | | FreeLB | | | FreeLB | 0.4614 | 0.6098 | 0.8194 | 0.9483 |
| **Combined-Model PerturbScorer** | | | | | | | | | |
| IMDB | BERT Textfooler | BERT & | IMDB | BERT Textfooler | BERT & | 0.4258 | 0.5601 | 0.5945 | 0.7634 |
| | | | | | | 0.4997 | 0.6561 | 0.7973 | 0.9377 |
| AG's News | Rand. Perturb Textfooler | FreeLB | AG's News | Rand. Perturb Textfooler | FreeLB | 0.3694 | 0.5034 | 0.615 | 0.8 |
| | | | | | | 0.4024 | 0.5424 | 0.8061 | 0.9414 |

Table 5: Through results of PerturbScorer generalization tests.

# Appendix

**Through Results of Generalization Experiments of the PerturbScorer** In Table 5, we list the thorough results which is the expansion of the results shown in Table 4. As shown, we can observe that the experimental results are consistent with the analysis based on the partial results in Table 4. We can build a PerturbScorer that can be used in cross-perturbation, datasets, and models as a general PerturbScorer to build the connection between discrete and continuous perturbations.