# OpenReview forum: "PerturbScore: Connecting Discrete and Continuous Perturbations in NLP"
_EMNLP/2023/Conference — EMNLP 2023 Findings_

### Official Review · Reviewer_Rjy5 · 2023-07-30

**Typos Grammar Style And Presentation Improvements:** 1. Use of mixed case styles
**Soundness:** 4

**Excitement:**

5: Transformative: This paper is likely to change its subfield or computational linguistics broadly. It should be considered for a best paper award. This paper changes the current understanding of some phenomenon, shows a widely held practice to be erroneous in someway, enables a promising direction of research for a (broad or narrow) topic, or creates an exciting new technique.

**Paper Topic And Main Contributions:**

Robustness of the neural NLP models have become an extremely important topic as these neural models are abundantly used in real world applications. Perturbation-based robustness study is popular among the deep learning researchers, specifically, in the computer vision community where the nature of perturbation is continuous.  Perturbation-based robustness study in NLP faces the challenge of handling discrete input leading to discrete perturbation. This paper aims at connecting discrete perturbation to continuous perturbation so that perturbation in the continuous domain is applicable in NLP. Primary contributions of the paper include 1) quantifying the correlation between discrete and continuous perturbation, 2) designing a regression-based model (viz, PerturbScore) to estimate the correlation.

**Questions For The Authors:**

1. Page 4, Column 2: the sentence  ‘..... obtain a data tuple [S, P(S), \epsilon] , which is the correlation’ is a bit ambiguous. Assuming ‘correlation’ refers  to ‘\epsioln’, is it a correlation between the discrete and continuous perturbations? Or to me it seems that ‘\epsilon’ is a continuous surrogate for the discrete perturbation P(S). Please clarify. This may be confused with another correlation measured in section 4.2 in the form of Kendal and Pearson Index. Also, the tuple [S, P(S), \epsilon] is not a correlation.

2. Page 6, Column 1: How does better resistance of models trained FreeLB method against discrete and continuous perturbation verify assumption 1 and ensure correlation between discrete and continuous perturbation? There seems to be a logical gap. Please clarify

3. Page 6: Column 1: Table 2 shows count of data tuples under different \epsilon ranges. This shows most of the discrete distributions correspond to lower continuous perturbation value ranges. Some of the data tuples have been discarded? The claim is as a small proportion of data tuples have been discarded, the method is able to find the ‘norm ball that satisfies Assumption 3’. What is the criteria for discarding the data tuples?  If it is manually selected then size of the set of discarded tuples may vary. Also, it is apparent why discarding a small proportion of tuples leads to satisfaction of Assumption 3.

4. Page 6: Column 1: There is no evidence in the paper that “Textfooler method generates more discrete perturbations …… and corresponding continuous perturbations require larger norm balls. Please specify if these experimental observations have been explicitly presented in any part of the paper.

5. The cross-dataset analysis in Appendix shows that there is no significant difference in the correlation between edit-distance and PerturbScorer. In many cases, the correlation is poorer in PerturbScorer. Though the situation improves when the datasets are combined.  How these results reflect on the generalizability of the model?

**Reasons To Accept:**

1. The paper studies one of the most important aspects of NLP research, i.e., measuring the robustness of the neural NLP models.
2. To deal with imperceptibility of the discrete perturbations and combinatorial explosion in formulating those perturbations, the paper advocates for applying continuous perturbation.
3. I really appreciate the two staged approach adopted by the authors: Firstly, mapping discrete perturbation to continuous perturbation with a set of non-trivial assumptions. This formalisation along with an optimization method the authors presented an elegant method for generating training examples each consisting of original input, discrete perturbation and equivalent continuous perturbation. Secondly, using those training instances, a regression model has been proposed to predict continuous prediction given a discrete perturbation.
4. The authors performed extensive experiments to validate the claims and showcase the usefulness of the method

**Reasons To Reject:**

1. Many claims are ambiguous and are not supported by strong logic. (see questions)
2. Given the results, I have doubt in the generalizability of the proposed model.

**Reproducibility:**

4: Could mostly reproduce the results, but there may be some variation because of sample variance or minor variations in their interpretation of the protocol or method.

**Reviewer Confidence:**

3: Pretty sure, but there's a chance I missed something. Although I have a good feel for this area in general, I did not carefully check the paper's details, e.g., the math, experimental design, or novelty.

---

> ### Author Rebuttal · Authors · 2023-08-26
>
> We sincerely appreciate your valuable comments!
>
>
> Question #1
>
> The tuple [S, P(S), \epsilon] is not a correlation, the tuple is the constructed connection between a pair of discrete and continuous perturbations. We hope to study the correlation between discrete and continuous perturbations by constructing a tuple of the discrete and continuous perturbation pair.
> We will correct this representation.
> In Sec. 4.2, the correlation is the metric to evaluate the closeness between metrics that measure discrete and continuous perturbation.
> By measuring the correlation coefficient index between A(P(S), S) and \epsilon, we can study the correlation between discrete and continuous perturbations.
> We will clarify these statements for readers to clearly understand our work.
> We appreciate your careful reading!
>
>
>
> Question #2
>
> The FreeLB method utilizes gradients to craft adversaries with virtual perturbations on the embedding space to train the model.
> Therefore, the FreeLB model is more robust to gradient-based perturbations.
> In Figure 1, we can see that the FreeLB model is more robust to both gradient and discrete perturbations, the slopes of FreeLB models are flatter than normal models.
> The tendency is similar, which verifies Assumption 1 that continuous and discrete perturbations have similar effects on target models.
>
> As for the statement, 'FreeLB results show that discrete and continuous perturbation can be correlated':
> We should be more cautious about such a statement. That is, the FreeLB model is more robust against both discrete and continuous perturbation, which might indicate that discrete and continuous perturbations can be connected.
>
> Actually, the reason why NLP adversarial training methods using gradients such as FreeLB can improve model robustness against discrete perturbations is because PerturbScore shows that we can build the correlation between discrete and continuous perturbations.
>
> Knowing that we can build such a correlation, we are very excited since now we can explain why FreeLB and later adversarial training methods work (PerturbScore can provide proof for these adversarial training methods).
>
> We will clarify these statements more clearly in the revision.
> Again, we are grateful for your careful reading!
>
>
> Question #3
>
> The discarded tuples are due to the rounding process, which is because of the \varepsilon selection.
> If \varepsilon is set to an extremely small value (reaching zero), then we will not discard any samples at all, but in this case, the generation process will be very long and the generated data will be massive.
> For the calculation trade-off, we set a certain \varepsilon not too large, therefore, there might be some mismatch between discrete and continuous perturbations.
> For example, if  \varepsilon is very large, different discrete perturbations will fall into the same continuous perturbation norm, and the correlation is not learnable anymore.
> Therefore, in between lines 326-327, Algorithm line 9, we set a threshold \phi to guarantee that the correlation is within a small range so that the correlation is learnable.
> It is an implementation trade-off, not a theoretical drawback.
>
>
> Question #4
>
> About Textfooler perturbations: Textfooler perturbations can cause more severe damage to neural models than random perturbations (which is illustrated in the Textfooler paper).
> Such a claim is intuitive since Textfooler iterates the substitutes to find perturbations that can harm neural models more, while random perturbations are not 'adversarial' against neural models.
> Therefore, in Table 2, it is reasonable that Textfooler perturbations have large norm-bound corresponding perturbations since they are stronger discrete perturbations.
>
> Question #5
>
> The PerturbScorer is relatively less effective in studying the cross-dataset prediction, that is mainly because the average length of IMDB dataset is over 200 words and the average length of AG's News is only around 40.
> Therefore, discrete perturbations are more different between these two datasets, therefore the PerturbScorer is less effective in generalizing the prediction.
>
> On the other hand, though the PerturbScorer is slightly lower than the edit distance, we at first did not expect that PerturbScorer could generalize to different datasets, perturbation types, or models at all.
> Therefore, we believe that the observation that PerturbScorer can obtain similar results in cross-dataset prediction with edit distance could show that PerturbScorer can be generalized in different scenarios.
> Based on this motivation, we test the combination of dataset, perturbation, and models, and results show that PerturbScorer can learn such correlation, indicating that we can build a very strong PerturbScorer with multiple models, different text types, and different perturbation types as a firm bridge between discrete and continuous perturbations.

---

### Official Review · Reviewer_WvbR · 2023-08-04

**Soundness:** 3

**Excitement:**

4: Strong: This paper deepens the understanding of some phenomenon or lowers the barriers to an existing research direction.

**Paper Topic And Main Contributions:**

This paper addresses the research question: can we find a connection between discrete perturbation and continuous perturbation to study the robustness of neural NLP models? They approach this question by quantifying the connections with the norm-bound of gradient-based perturbations and training a regression model PerturbScorer to predict the strength of the connection (correlation). The resulting PerturbScorer can be useful for better understanding robust training methods of NLP models and transferring the robustness theories studied in the computer vision field to NLP.

**Questions For The Authors:**

1. The paper explores how to approximate a discrete perturbation with a continuous one. Is it possible to do the other way round, i.e. finding the corresponding discrete perturbation of a continuous one? This could make the two types of perturbations more connected.

2. Do you use the same model (e.g. BERT) for perturbation dataset generation (sec 3.4) and perturbscoer (sec 3.5)? What would the correlation results be like if you use different types of models?

**Reasons To Accept:**

1. This paper looks into an interesting problem of finding a continuous perturbation as a proxy for discrete perturbation, which provides a new angle to study the robustness of NLP models through the lens of continuous space.

2. Experiment results of PerturbScorer look promising, indicating that the generated perturbation dataset and the trained model are of good quality.

3. The method is carefully designed and well-explained in Sec 3.

**Reasons To Reject:**

1. Limited numbers of models and datasets/tasks are studied in this paper. I hope to see similar results on more model architectures (including a simpler RNN model) and tasks to better support the generalization ability of PerturbScorer.

2. I wish to see a concrete demonstration of how the PerturbScorer can be applied to broader scenarios such as studying the robustness of NLP models in continuous space, measuring sentence differences, etc.

3. The validity of the proposed method would be better supported by a theoretical proof of when the norm-bound would exist (the continuous perturbation effect is way bigger than discrete ones, line 342-345).

**Reproducibility:**

4: Could mostly reproduce the results, but there may be some variation because of sample variance or minor variations in their interpretation of the protocol or method.

**Reviewer Confidence:**

3: Pretty sure, but there's a chance I missed something. Although I have a good feel for this area in general, I did not carefully check the paper's details, e.g., the math, experimental design, or novelty.

**Typos Grammar Style And Presentation Improvements:**

The paper seems to miss a brief introduction to the FreeLB method appeared in Fig 1 & Table 2. Can you add a few sentences to explain it?

---

> ### Author Rebuttal · Authors · 2023-08-26
>
> We sincerely appreciate your valuable comments!
>
>
> Question #1
>
> We have considered studying the correlation the other way around and we think this is a very interesting idea.
> One major obstacle is that we round continuous perturbations using norm bound without further studying the direction of the continuous perturbations.
> If we find discrete perturbations from a continuous perturbation, the searching space would be extremely large.
> Such an idea could be possible with some very powerful generative models, therefore, we think that building such a correlation by fine-tuning GPT-4 or at least fine-tuning LLaMA might be a plausible solution, which could be a promising future work of PerturbScore.
>
> Question #2
> We use BERT-BASE to train the PerturbScorer and the perturb dataset generation includes FreeLB trained BERT in IMDB and AG's News dataset and normal fine-tuned BERT in IMDB and AG's News dataset.
> As seen, the FreeLB-trained model and the normal fine-tuned model generated perturb-dataset show similar results using BERT-trained PerturbScorer, therefore we can conclude that the PerturbScorer does not require the same base model on which the perturbation is constructed based.
>
> Still, we appreciate such a question and we will add some other models such as RoBERTa or LLaMA models to train the PerturbScorer to better study the correlation between discrete and continuous perturbations.
>
>
>
>
> Weakness #1
>
> We will add more models and datasets to the revision. We will add simple models including RNNs and stronger models including LLaMA to study the perturbation correlation.
> Also, we will add more datasets including classification datasets such as SST-2 and Yelp, and NLI datasets including SNLI and MNLI.
>
> Weakness #2
>
> We will add an additional experiment using our proposed PerturbScore as a metric to evaluate the perturbations added in an adversarial attack process (such as Textfooler attack or BERT-Attack).
> Plus, we will add an experiment to use PerturbScore as a scorer to study whether it can be used in measuring sentence similarities.
> We are hoping that by adding these experiments, we will be able to strengthen the importance of building PerturbScore in studying the correlation between continuous and discrete perturbations.
>
> Weakness #3
>
> As for the question about discarding the mismatch between continuous perturbations and discrete perturbations, we can prove that the norm bound does exist. We discard some cases simply for the implementation rounding.
>  That is, with no perturbation, the model is not changed at all, and with as many perturbations as possible, the model will be shifted to a random prediction. Within this range, there will always be a norm-bound to match the discrete perturbation.
> We discard some of the mismatch cases because the small interval $\varepsilon$ is not an extremely small value.
> If the interval is an extremely small value (reaching 0), there will always be a matching pair of discrete and continuous perturbations, but the calculation will be massive therefore we set the value as a trade-off in the implementation.
> We will clarify this in the paper to avoid this misunderstanding.
>
>
> Introduction of the FreeLB method
>
> The FreeLB method is a simple modification of the PGD method used to craft gradient-based adversaries.
> The adversarial training process is similar to Algorithm 1. The algorithm will first calculate gradients and then add the gradients to the input embedding as a virtual adversary to improve model robustness.
> We use the gradients update process to find the correlation between discrete and continuous perturbations. Therefore, we initially believe that the adversarial-trained model should follow Assumption 2 better as it is more robust against gradient-based perturbations.
> Results in Table 3 show that the FreeLB-trained model has a tighter correlation between discrete and continuous perturbations, which is intuitive given the adversarial training process helps build a tighter bound of continuous perturbations.

---

### Official Review · Reviewer_pixE · 2023-08-04

**Typos Grammar Style And Presentation Improvements:** 1. I found it difficult to read Secti…
**Soundness:** 3

**Excitement:**

4: Strong: This paper deepens the understanding of some phenomenon or lowers the barriers to an existing research direction.

**Paper Topic And Main Contributions:**

In this paper, the authors study the problem of model robustness in NLP. Specifically, they seek to study the relationship between continuous and discrete perturbations in NLP models. The authors motivate this study by noting tha most robustness work in NLP focuses on discrete perturbations. However, this can be problematic as discrete perturbations are much more costly than their continuous counterparts. The authors first assume that there exists a connection between discrete and continuous perturbations, and using that design a method finds the proper norm bound between a continuous and discrete perturbation. They utilize this method to construct their own pertuber. They utilize this to conduct various studies comparing the two types of perturbations and the effect on model performance.

Post-Rebuttal Update: I appreciate the detailed response. I find most of the responses satisfactory. I've raised my excitement from a 3 &rarr; 4

**Questions For The Authors:**

1. What are the axes for the the plots in figure 1? I'd recommend including axis names in the future for better readability.

2. Could you re-explain the significance of the findings from Section 4.4? See my comments in the weaknesses section.

3. Could you expound on some potential applications of this work? Ideally including a small section in the paper about this would be helpful.

**Reasons To Accept:**

1. The goal of this study is well-motivated and clear.

2. The authors do a strong job of supporting the assumptions made in Section 2 through empirical study.

3. The empirical results are comprehensive. Multiple datasets and methods are used.

**Reasons To Reject:**

1. More care can be taken in explaining the equation in Section 3. It was not obvious at first as to what exactly they should be measuring. An explanation given before or after each equation, explaining the purpose and intuition would be helpful.

2. I don't find the results in presented in Section 4.4 to be very convincing. First, based on the results the authors state that the relationship between the discrete and continuous perturbations is weak. They conclude that discrete perturbations can't properly attack models. However, earlier they use Figure 1 to show that the continuous and discrete perturbations have a similar effect on model performance. It's hard to square these two assertions. Secondly, they note that PerturbScorer has a strong correlation with the continuous pertubations. I feel like this isn't surprising at all as PerturbScorer utilizes $\epsilon$.

**Reproducibility:**

4: Could mostly reproduce the results, but there may be some variation because of sample variance or minor variations in their interpretation of the protocol or method.

**Reviewer Confidence:**

3: Pretty sure, but there's a chance I missed something. Although I have a good feel for this area in general, I did not carefully check the paper's details, e.g., the math, experimental design, or novelty.

---

> ### Author Rebuttal · Authors · 2023-08-26
>
> We sincerely appreciate your valuable comments!
>
> About Weakness #1:
>
> In Equation 1, we hope to find the minimum norm bound that is equal to the output shift caused by the discrete perturbation S, and in Algorithm 1 we iterate gradient-based adversaries of different norm bounds to match the discrete perturbation. By using the gradient-based adversary, which is the min-max problem in standard adversarial training, we are able to use norm-bound to represent continuous perturbation.
> Then we can use Equation 1 to connect discrete perturbation S and continuous perturbation \delta.
> We will add these illustrations in the revision to avoid misunderstanding.
> We appreciate your careful reading!
>
>
> About Weakness #2:
>
> Please see Question #2.
>
> About Question #1:
>
> In Figure 1, the y-axis is the cosine similarity of perturbed text and original text.  In subfigure (a) and (b) the x-axis is the edit distance and in (c) and (d) the x-axis is the norm-ball range.
> We will add the name clearly  (we illustrate them in captions first).
> The figure verifies Assumption 1 that both discrete and continuous perturbations can cause more muscular damage to the model when the perturbations increase.
>
> About Question #2:
>
> "The relationship between discrete and continuous perturbations is weak" is to measure discrete perturbations using metrics such as edit-distance or BERTScore. Our proposed PerturbScore can build the relationship between discrete and continuous perturbations.
>
> In Figure 1, the similar effect is to consider perturbations separately, that is, stronger perturbations indicate a strong attack effect.
>
> For example, larger norm-bound perturbations can cause more damage to the model, and larger discrete perturbations can cause more damage to the model as well.
> But larger norm-bounds DO NOT equal large discrete perturbations.
>
> Figure 1 is not a case-by-case figure, therefore, cannot show the mismatch between discrete and continuous perturbations.
> We show the case-by-case mismatch between discrete and continuous perturbations in Sec. 4.4 (Table 3)
> As seen, metrics such as edit distance and BERTScore cannot measure discrete perturbations, while PerturbScore shows a close correlation between discrete and continuous perturbations.
>
> Therefore, we aim to build the connection between discrete and continuous perturbations, and we can see that our proposed PerturbScore can measure how strong the discrete perturbations are, while edit distances can not.
> PerturbScore is to learn the correlation between discrete and continuous perturbations. Our experiment shows that it is POSSIBLE to utilize $\epsilon$ to study such correlation.
> Previous works do not show that a continuous bound $\epsilon$ can represent a discrete perturbation, which is the major contribution of PerturbScore.
>
> About Question #3:
>
> A very straightforward application of PerturbScore is to quantify discrete perturbations.
> For example, in protecting model robustness against adversarial attacks, a strong PerturbScore can judge whether crafted perturbations can harm the model and protect the model.
> Plus, as discussed in [FreeLB: Enhanced Adversarial Training for Natural Language Understanding(Zhu et al. 2019)], [Token-aware virtual adversarial training in natural language understanding (Li and Qiu, 2021)], [Searching for an Effective Defender: Benchmarking Defense against Adversarial Word Substitution], adversarial training with virtual perturbations can help improve robustness against discrete attacks.
> These papers do not give any explanation of why continuous perturbations can help improve robustness, with PerturbScorer, we can prove that there can be a correlation between continuous and discrete perturbations, providing proof for studying those adversarial training methods to improve model robustness.
> Also, the PerturbScore can measure the continuous perturbations, which can be generalized to a sentence similarity checker similar to BERTScore.
> On the other hand, PerturbScore can blur the line between discrete and continuous space, which can help build multi-modal models based on the correlation between discrete and continuous perturbations (e.g. help building negative samples for cross-modal general large models)
>
>
>
> About Presentation Improvements:
>
> We will re-construct the paper to make it more clear for readers, we appreciate your comments!

---

### Meta-Review · Area_Chair_5Gr8 · 2023-09-23

**Recommendation:** 1

**Metareview:**

The paper tries to connect continuous and discrete perturbations. Particularly, this paper suggests an algorithm to find a continuous perturbation given discrete perturbation within some error bound and further suggests a regressor (PerturbScorer) that emits matching continuous perturbation’s norm-bound given original sentence S, and discrete perturbation P(S).

The responses from the reviewers are somewhat unanimous. This paper brings an interesting and novel direction (and thus high excitement scores), however, there are concerns regarding clarity, generalizability, and the method’s application in a broader perspective (resulting in insufficient soundness score).

I too agree that this is an interesting direction, but also agree with other reviewers that the manuscript lacks clarity and that the broader application should be addressed further. Authors do respond that this can be useful in multi-modal settings and help explain previous papers. However, it seems that there is a nontrivial step to show this usefulness on top of what the paper already shows. To be more precise with some examples, can this algorithm tell why some continuous perturbations worked and some did not? Addressing the concerns reviewers brought up and resolving ambiguities will significantly enhance the paper's value to the NLP community.

(Some more detailed examples that the paper can improve on)

- (Figure 1) I was not able to understand Figure 1 at all by just reading the paper. I was only able to understand this after looking back and forth at the comments the authors made in the rebuttal. Even then, the interpretation of it was not very straightforward.
- Correlation is somewhat overused. For example, the paper studies the correlation between the effect of discrete and continuous perturbations and also names the output of the PerturbScorer to be outputting correlation.
- Authors claim that this method can help multi-modal setups. However, they also respond in their rebuttal that PerturbScorer is not very effective in generalizing prediction when overall statistics (word lengths) are different. This result was on a very similar task but with different sentence-length statistics. In a truly multi-modal setup, would the proposed method be helpful? + Why is it very useful to connect two modalities in the first place was not very clear to me in the paper. Delivering those motivations clearly could also help enhance the method’s impact in the community.
- As many reviewers pointed out, some of the statements are strong but unwarranted in a theoretical manner. e.g.) "proving" is a strong word in 418-423: "...proving that we can successfully find norm balls ...". How about changing this to a softer sentence? For example, one could say "Our experiments display, that under XYZ kind of perturbations, the proposed method can successfully find matching continuous perturbations.."

---

### Decision · Program_Chairs · 2023-10-07

**Decision:**

Accept-Findings

**Comment:**

The paper tries to connect continuous and discrete perturbations. Particularly, this paper suggests an algorithm to find a continuous perturbation given discrete perturbation within some error bound and further suggests a regressor (PerturbScorer) that emits matching continuous perturbation’s norm-bound given original sentence S, and discrete perturbation P(S).

The responses from the reviewers are somewhat unanimous. This paper brings an interesting and novel direction (and thus high excitement scores), however, there are concerns regarding clarity, generalizability, and the method’s application in a broader perspective (resulting in insufficient soundness score).

I too agree that this is an interesting direction, but also agree with other reviewers that the manuscript lacks clarity and that the broader application should be addressed further. Authors do respond that this can be useful in multi-modal settings and help explain previous papers. However, it seems that there is a nontrivial step to show this usefulness on top of what the paper already shows. To be more precise with some examples, can this algorithm tell why some continuous perturbations worked and some did not? Addressing the concerns reviewers brought up and resolving ambiguities will significantly enhance the paper's value to the NLP community.

(Some more detailed examples that the paper can improve on)

- (Figure 1) I was not able to understand Figure 1 at all by just reading the paper. I was only able to understand this after looking back and forth at the comments the authors made in the rebuttal. Even then, the interpretation of it was not very straightforward.
- Correlation is somewhat overused. For example, the paper studies the correlation between the effect of discrete and continuous perturbations and also names the output of the PerturbScorer to be outputting correlation.
- Authors claim that this method can help multi-modal setups. However, they also respond in their rebuttal that PerturbScorer is not very effective in generalizing prediction when overall statistics (word lengths) are different. This result was on a very similar task but with different sentence-length statistics. In a truly multi-modal setup, would the proposed method be helpful? + Why is it very useful to connect two modalities in the first place was not very clear to me in the paper. Delivering those motivations clearly could also help enhance the method’s impact in the community.
- As many reviewers pointed out, some of the statements are strong but unwarranted in a theoretical manner. e.g.) "proving" is a strong word in 418-423: "...proving that we can successfully find norm balls ...". How about changing this to a softer sentence? For example, one could say "Our experiments display, that under XYZ kind of perturbations, the proposed method can successfully find matching continuous perturbations.."